# Thermodynamics Beyond Molecules: Statistical Thermodynamics of Probability Distributions

**DOI:** 10.3390/e21090890

**Published:** 2019-09-13

**Authors:** Themis Matsoukas

**Affiliations:** Department of Chemical Engineering, Pennsylvania State University, University Park, PA 16802, USA; txm11@psu.edu; Tel.: +1-814-863-2002

**Keywords:** statistical thermodynamics, statistical mechanics, biased sampling, most probable distribution

## Abstract

Statistical thermodynamics has a universal appeal that extends beyond molecular systems, and yet, as its tools are being transplanted to fields outside physics, the fundamental question, what is thermodynamics, has remained unanswered. We answer this question here. Generalized statistical thermodynamics is a variational calculus of probability distributions. It is independent of physical hypotheses but provides the means to incorporate our knowledge, assumptions and physical models about a stochastic processes that gives rise to the probability in question. We derive the familiar calculus of thermodynamics via a probabilistic argument that makes no reference to physics. At the heart of the theory is a space of distributions and a special functional that assigns probabilities to this space. The maximization of this functional generates the mathematical network of thermodynamic relationship. We obtain statistical mechanics as a special case and make contact with Information Theory and Bayesian inference.

## 1. Introduction

What is thermodynamics? The question, so central to physics, has been asked numerous times and has been given nearly as many different answers. To quote just a few: thermodynamics is “the branch of science concerned with the relations between heat and other forms of energy involved in physical and chemical processes” [1]; “the study of the restrictions on the possible properties of matter that follow from the symmetry properties of the fundamental laws of physics” [2]; “concerned with the relationships between certain macroscopic properties of a system in equilibrium” [3]; “a phenomenological theory of matter” [4]. Such statements, while strictly true, focus on aspects that are far too narrow to converge to a definition of sufficient generality as to *what* to call thermodynamics or *how* to carry it outside physics. And yet, since Gibbs [5], Shannon [6] and Jaynes [7] drew quantitative connections between entropy and probability distributions, thermodynamics has been spreading to new fields. The tools of statistical thermodynamics are now used in network theory [8], ecology [9], epidemics [10], neuroscience [11], financial markets [12], and in the study of complexity in general. What motivates the impulse to apply thermodynamics to such vastly diverse problems? Is thermodynamics even applicable outside classical or quantum mechanical systems? And if so, what is the scope of its applicability?

Here we answer these fundamental questions: Statistical thermodynamics is variational calculus applied to probability distributions and by extension to stochastic processes in general; it is independent of physical hypotheses but provides the means to incorporate our knowledge and model assumptions about the particular problem. The fundamental ensemble is a space of probability distributions sampled via a bias functional. The maximization of this functional expresses a distribution—any distribution—via a set of parameters (microcanonical partition function, canonical partition function and generalized temperature) that are connected through a set of mathematical relationships that we recognize as the familiar equations of thermodynamic. Entropy and the second law have simple interpretations in this theory. We obtain statistical mechanics as a special case and make contact with Information Theory and Bayesian inference.

## 2. The Calculus of Statistical Thermodynamics

Before we derive a theory of generalized thermodynamics we review the key elements of the standard thermodynamic calculus. The central quantity of interest in statistical thermodynamics is the probability of microstate. For a system of *N* particles in volume *V* and temperature *T* this probability is given by the exponential (canonical) distribution,
(1)$$\mathrm{Prob}(\mathrm{microstate}\;i)=\frac{e^{-\beta E_{i}}}{Q},$$
where *Q* is the canonical partition function, $E_{i}$ is the energy of microstate, $\beta =1/k_{B}T$ and $k_{B}$ is Boltzmann’s constant. The corresponding probability to find the system in a microstate with energy *E* is obtained by summing all microstates with fixed energy *E* and is given by
(2)$$\mathrm{Prob}\left(E\right)=\Omega \frac{e^{-\beta E}}{Q},$$
where Ω is the microcanonical partition function, also equal to the number of microstates with energy *E*, volume *V* and number of particles *N*. The mean energy $\bar{E}$ and the parameters Ω, *Q* and β that appear in Equation (Equation 2) are interrelated: (3)$$\begin{array}{c} \log\Omega =\beta \bar{E}+\log Q, \end{array}$$(4)$$\begin{array}{c} \beta =\frac{\partial \log\Omega }{\partial \bar{E}}, \end{array}$$(5)$$\begin{array}{c} \bar{E}=-\frac{\partial \log Q}{\partial \beta }, \end{array}$$(6)$$\begin{array}{c} \frac{\partial^{2}\log\Omega }{\partial {\bar{E}}^{2}}\leq 0. \end{array}$$

Equations (Equation 3) and (Equation 4) establish that logΩ(E,V,T) and logQ(β,V,N) are Legendre pairs; Equation (Equation 6) states that logΩ is concave. In addition, any probability distribution $p_{i}$ that could be assigned to microstate *i* under fixed $\left(\bar{E},V,N\right)$ satisfies the inequality,
(7)$$-\sum_{i}p_{i}\log p_{i}\leq \log\Omega ,$$
with the equal sign only for the canonical distribution in Equation (Equation 1). This inequality is the statistical expression of the second law. If we identify $k_{B}\log\Omega$ with entropy and −(logQ)/β with free energy Equations (Equation 3)–(Equation 6) represent the familiar relationships of classical thermodynamics. Along with Equations (Equation 2) and (Equation 7), which provide the probabilistic context, the above set comprises the core relationships of statistical thermodynamics. The physical assumptions and postulates that produce these results can be found in any standard textbook (for example [3]). We will now show that this network of mathematical relationships arises naturally via a probabilistic construction that makes no reference to physics and endows any probability distribution f(x), x≥0 with the thermodynamic relationships shown here.

## 3. Theory

### 3.1. Random Sampling

Consider the continuous probability distribution $h_{0}\left(x\right)\geq 0$, $x\in (x_{a},x_{b})$, normalized to unit area. We define a discrete grid $x_{i}=x_{a}+\left(i-1\right)\Delta$ with $\Delta =(x_{b}-x_{a})/K$, i=1,2⋯K+1, such that the probability to sample a value of *x* in the *i*th interval is
(8)$$p_{i}=h_{0}\left(x_{i}\right)\Delta ,$$
if Δ is sufficiently small. We sample *N* values from $h_{0}$ and construct the frequency distribution $n=(n_{1},n_{2},\cdots )$, where $n_{i}$ is the number of sampled values that lie in the *i*th interval. The probability to observe distribution n in a random sample of size *N* is
(9)$$P\left(n|p,N\right)=N!\prod_{i}\frac{p_{i}^{n_{i}}}{n_{i}!},$$
and its logarithm is
(10)$$\log P\left(n|p,N\right)=-\sum_{i}n_{i}\log\frac{n_{i}}{p_{i}N}+O\left(\log N\right),$$
where $p=(p_{1},p_{2}\cdots )$. We define $h\left(x_{i}\right)=n_{i}/N\Delta$ and take the limit Δ→0, N→∞ in Equation (Equation 10). We then have $P\left(n|p,N\right)\to \delta P\left(h|h_{0},N\right)$ and
(11)$$\frac{\log\delta P(h|h_{0},N)}{N}=-\int h\left(x\right)\log\frac{h(x)}{h_{0}\left(x\right)}dx≐-D(h∥h_{0}),$$
where $\delta P(h|h_{0},N)$ is the probability to sample region (h,h+δh) in the continuous space of distributions, while taking a random sample of size *N* from $h_{0}$ (all integrals are understood to be taken over the domain of $h_{0}$). Any probability distribution h(x) defined in the domain of $h_{0}$ may materialize in a random sample taken from $h_{0}$. Clearly, the most probable distribution in this space is $h_{0}$ and indeed $h_{0}$ maximizes Equation (Equation 11). For all other distributions we must have $\delta P\left(h|h_{0},N\right)\leq \delta P\left(h_{0}|h_{0},N\right)=1$, or
(12)$$D(h∥h_{0})\geq 0,$$
with the equal sign only for $h=h_{0}$. The probability in the limit N→∞ to obtain $h_{0}$ relative to the probability to obtain any other distribution is

(13)$$\frac{\delta P(h_{0}|h_{0},N)}{\delta P(h|h_{0},N)}=e^{ND(h∥h_{0})}\to \infty .$$

Accordingly, $h_{0}$ is overwhelmingly more probable than any other distribution in its domain.

These results make contact with a broader mathematical literature. The quantity $D(h∥h_{0})$ in Equation (Equation 11) is the relative entropy (Kullback-Leibler divergence) of distribution *h* with respect to $h_{0}$, and plays an important role in Information Theory [13,14,15]; Equation (Equation 12) is the Gibbs inequality, a well known property of relative entropy; the relationship between relative entropy and the probability of a sample drawn from $h_{0}$ is a known result in the theory of large deviations [16]. The key point we take from these results is that the process of sampling distribution $h_{0}$ establishes a probability space of distributions with the same domain as $h_{0}$—these are the distributions obtained as samples. The Gibbs inequality states the elementary fact that the most probable distribution in this space is $h_{0}$. We will now generalize this probability space and the Gibbs inequality.

### 3.2. Biased Sampling

Random sampling always converges to the distribution from which the sample is taken; the probability of all other distributions vanishes as N→∞. We now modify the sampling process in order to obtain some different limiting distribution $h^{*}$ while still sampling from $h_{0}$. We do this by applying a bias, such that a random sample of size *N* from $h_{0}$ is accepted with probability proportional to W[Nh], where Nh is the frequency distribution of the sample and *W* is a functional with the homogeneous property logW[Nh]=NlogW[h]. We require homogeneity so that the limiting distribution is independent of *N* when N→∞. By virtue of homogeneity, logW is written as
(14)logW[h]=∫h(x)logw(x;h)dx,
where logw(x;h) is the variational derivative of logW[h] with respect to *h*. The probability to obtain a sample with distribution *h* under this biased sampling is
(15)$$P\left(h|p,W,N\right)=\frac{W[Nh_{i}]}{r^{N}}\left(N!\prod_{i}\frac{p_{i}^{n_{1}}}{n_{i}!}\right),$$
where $r^{N}$ is a normalizing constant; the logarithm of this probability in the continuous limit is

(16)$$\frac{\log\delta P(h|h_{0},W,N)}{N}=-\int h\left(x\right)\log\frac{h(x)}{w\left(x;h\right)h_{0}\left(x\right)}dx-\log r.$$

We define the probability functional
(17)$$\log\varrho \left[h|h_{0},W\right]≐-\int h\left(x\right)\log\frac{h(x)}{w\left(x;h\right)h_{0}\left(x\right)}dx-\log r,$$
so that the probability to observe a distribution within (h,h+δh) in a biased sample taken from $h_{0}$ is $\delta P\left(h|h_{0},N\right)=\varrho^{N}\left[h|h_{0},W\right]$. The ratio of the probability to sample the most probable distribution $h^{*}$ relative to that for any other distribution in the continuous limit is

(18)$$\frac{\delta P(h^{*}|h_{0},W,N)}{\delta P(h|h_{0},W,N)}={\left(\frac{\varrho [h^{*}|h_{0},W]}{\varrho [h|h_{0},W]}\right)}^{N}\to \infty .$$

As in random sampling, the most probable distribution is overwhelmingly more probable than any other feasible distribution. Then we must have
(19)$$\varrho [h|h_{0},W]\leq 1,$$
with the equal sign only for the most probable distribution $h^{*}$. This distribution is (see Appendix A).

(20)$$h^{*}\left(x\right)=w\left(x;h^{*}\right)\frac{h_{0}\left(x\right)}{r},$$
with *r* determined by normalization. If we choose $w\left(x;h\right)=f\left(x\right)/h_{0}\left(x\right)$, where *f* is any other normalized distribution in the domain of $h_{0}$, we obtain $h^{*}=f$. Therefore, a suitable bias can always be constructed such that *any* distribution in the domain of $h_{0}$ may be obtained as the most probable distribution by biased sampling of $h_{0}$; conversely, any distribution $h_{0}$ may be used to generate a sample of any other distribution *f* over the same domain by biased sampling.

### 3.3. Canonical Sampling

We now choose the generating distribution $h_{0}$ to be the normalized exponential distribution with parameter β,
(21)$$h_{0}\left(x\right)=\beta e^{-\beta x};\;0\leq x<\infty ,$$
and write the probability functional ϱ in Equation (Equation 17) as
(22)$$\log\varrho \left[h|W,\beta \right]=-\int h\left(x\right)\log\frac{h(x)}{w(x;h)}dx-\beta \bar{x}-\log q,$$
where q=r/β and $\bar{x}$ is the mean of h(x). We call this probability space *canonical*. The probability of *h* is $\varrho^{N}\left[h|W,\beta \right]$ and by the same argument that led to Equation (Equation 19) we now have

(23)ϱ[h|W,β]≤1.

The equal sign defines the most probable distribution $h^{*}$; this distribution is

(24)$$h^{*}\left(x\right)=w\left(x;h^{*}\right)\frac{e^{-\beta x}}{q}.$$

The parameter *q* is fixed by the normalization condition and satisfies

(25)$$\bar{x}=-\frac{d\log q}{d\beta }.$$

(Details are given in Appendix A).

### 3.4. Microcanonical Sampling

Next we define the *microcanonical* space as the subset of distributions with fixed mean $\bar{x}$. The generating distribution is again the exponential function, which we now write as
(26)$$h_{0}\left(x\right)=\frac{e^{-x/\bar{x}}}{\bar{x}},$$
with $\bar{x}$ fixed. The probability to observe distribution *h* while sampling $h_{0}$ is still given by Equation (Equation 16) except that *r* is replaced with a new normalizing factor $r^{\prime }$. We define the microcanonical probability functional
(27)$$\log\varrho \left[h|W,\bar{x}\right]=-\int h\left(x\right)\log\frac{h(x)}{w(x;h)}dx-\log\omega ,$$
with $\log\omega =1+\log\bar{x}+\log r^{\prime }$ and write the probability of *h* as $\varrho^{N}\left[h|W;\bar{x}\right]$. The argument that produced Equations (Equation 19) and (Equation 23) now gives

(28)$$\varrho [h|W,\bar{x}]\leq 1.$$

This functional is maximized by the same distribution $h^{*}$ that maximizes the canonical functional, Equation (Equation 24), except that both *q* and β are now Lagrange multipliers and are fixed by normalization and by the known mean $\bar{x}$. As in the canonical case, $h^{*}$ is overwhelmingly more probable than any other distribution in the microcanonical space and its mean satisfies Equation (Equation 25). We insert Equation (Equation 24) into (Equation 28) to obtain
(29)$$\log\omega =S\left[h^{*}\right]+\log W\left[h^{*}\right],$$
where $S[h^{*}]$ is the Gibbs–Shannon entropy of the most probable distribution,

(30)$$S\left[h^{*}\right]=-\int_{0}^{\infty }h^{*}\left(x\right)\log h^{*}\left(x\right)dx.$$

Substituting Equation (Equation 24) for $h^{*}$ in (Equation 29) we obtain a relationship between ω, β, *q* and $\bar{x}$:(31)$$\log\omega =\beta \bar{x}+\log q.$$

In combination with Equation (Equation 25), this result defines $\log\omega (\bar{x})$ as the Legendre transformation of q(β) with respect to β. By the reciprocal property of the transformation we then have

(32)$$\beta =\frac{d\log\omega }{d\bar{x}}.$$

Given Equation (Equation 31), the canonical probability functional in Equation (Equation 22) and the microcanonical functional in Equation (Equation 27) are seen to be the same. The difference is that in canonical maximization $\bar{x}$ is a floating parameter, whereas in the microcanonical maximization it is held constant. Both functionals are maximized by the same distribution and have the same β, *q*, ω at same $\bar{x}$: the two ensembles are equivalent. Finally, the maximization of the microcanonical functional implies that $\varrho [h;W,\bar{x}]$ is a concave functional in *h*. It follows that logω is a concave function of $\bar{x}$, therefore we must have

(33)$$\frac{d^{2}\log\omega }{d{\bar{x}}^{2}}=\frac{d\beta }{d\bar{x}}\leq 0.$$

The details are shown in Appendix A.

## 4. Generalized Statistical Thermodynamics

These results can be summarized in the form of the following theorem:

**Theorem** **1.**
*Given normalized distribution f(x), x≥0, with mean $\bar{x}$, it is possible to construct a functional W such that:*

*(a) All distributions h(x), x≥0, with mean $\bar{x}$ satisfy the inequality*
(34)$$\log W\left[h\right]-\int_{0}^{\infty }h\left(x\right)\log h\left(x\right)dx\leq \log\omega$$
*with the equal sign only for h=f, a condition that defines ω;*

*(b) f can be expressed in canonical form as*
(35)$$f\left(x\right)=w\left(x\right)\frac{e^{-\beta x}}{q},$$
*where logw is the variational derivative of logW[f]; and*

*(c) parameters $\bar{x}$, β, q and ω satisfy*
(36)$$\begin{array}{c} \bar{x}=-\frac{d\log q}{d\beta }, \end{array}$$
(37)$$\begin{array}{c} \beta =\frac{d\log\omega }{d\bar{x}}, \end{array}$$
(38)$$\begin{array}{c} \log\omega =\beta \bar{x}+\log q, \end{array}$$
(39)$$\begin{array}{c} \frac{d^{2}\log\omega }{d{\bar{x}}^{2}}\leq 0. \end{array}$$


The existence of *W* is established by the fact that the functional
(40)$$\log W\left[h\right]=\int_{0}^{\infty }h\left(x\right)\log f\left(x\right)dx,$$
satisfies the theorem. This is a linear functional whose derivative is logf for all *h*. More generally, any homogeneous functional logW[h] of degree 1, linear or non-linear, whose derivative at h=f is given by
(41)$${\left.\frac{\delta \log W[h]}{\delta h}\right|}_{h=f}=\log f\left(x\right)+a_{0}+a_{1}x≐\log w\left(x\right),$$
where $a_{0}$ and $a_{1}$ satisfy
(42)$$\frac{da_{0}}{da_{1}}=-\bar{x},$$
but are otherwise arbitrary, also satisfies the theorem. The inequality in Equation (Equation 39) follows from the concave requirement that ensures the maximization of Equation (Equation 34).

We recognize Equation (Equation 35) as the canonical distribution of statistical mechanics, Equations (Equation 36)–(Equation 38) and (Equation 33), which relate its parameters, as the core set of thermodynamic relationships, and Equation (Equation 34) as the inequality of the second law. The probabilistic interpretation is that any distribution *f* may be obtained as the most probable distribution under a probability measure defined via a suitable functional *W*. Whereas in statistical mechanics the central stochastic variable is the mechanical microstate, in generalized thermodynamics it is the probability distribution itself. Thermodynamics may be condensed into the microcanonical inequality in Equation (Equation 34), a generalized expression of the second law that defines the most probable distribution in the microcanonical space. All relationships between ω (microcanonical partition function), *q* (canonical partition function), β (generalized inverse temperature) and $\bar{x}$ follow from the maximization of this inequality and have equivalents in familiar thermodynamics. The derivatives dlogq/dβ and $d\log\omega /d\bar{x}$ in Equations (Equation 36) and (Equation 37) may be viewed as equations of change along a path (“process”) in the space of distributions under fixed bias *W*. This path is described parametrically in terms of $\bar{x}$ and represents a nonstationary stochastic process. We call this process *quasistatic*—a continuous path of distributions that maximize locally the thermodynamic functional.

### 4.1. Contact with Statistical Mechanics

The obvious way to make contact with statistical mechanics is to take *f* to be the probability of microstate at fixed temperature, volume and number of particles. The postulate of equal a priori probabilities fixes the selection functional and its derivative to W=w=1; if we identify *x* as the energy $E_{i}$ of microstate *i*, β as $1/k_{B}T$, *q* as the thermodynamic canonical partition function, ω as the thermodynamic microcanonical partition function, Equations (Equation 24)–(Equation 33) map to standard thermodynamic relationships. From Equation (Equation 29) we obtain $\varrho =e^{S[h]}/\omega$: the canonical probability *f* maximizes entropy and thus we obtain the second law.

This is not the only way to establish contact with statistical mechanics. We may choose *f* to be some other probability distribution, for example, the probability to find a *macroscopic* system of fixed (T,V,N) at energy *E*. We write the energy distribution in the form of Equation (Equation 24) with *w*, β and *q* to be determined. From Equations (Equation 25), (Equation 31) and (Equation 32) with $\bar{x}=\bar{E}$ we make the identifications $\beta \to 1/k_{B}T$, $\log q\to -F/k_{B}T$ (free energy), logω→ thermodynamic entropy. To identify *w* we require input from physics and this comes via the observation that the probability density of macroscopic energy *E* is asymptotically a Dirac delta function at $E=\bar{E}$. Then S[f]=0 (this is the entropy of the energy distribution, not to be confused with thermodynamic entropy). From Equations (Equation 14) and (Equation 29) we find logW[f]=logw(x;f)=logω, and conclude that logω is the thermodynamic entropy. This establishes correspondence between generalized thermodynamics and macroscopic (classical) thermodynamics. If we further postulate, again motivated by physics, that w(E) is the number of microstates under fixed volume and number of particles, we establish the microscopic connection. Since f(E) is proportional to the number of microstates with energy *E* and individual microstates are unobservable, we may as well ascribe equal probability to all microstates. Thus we recover the postulate of equal a priori probabilities (statistical thermodynamics). Finally, by adopting a physical model of microstate, classical, quantum or other, we obtain classical statistical mechanics, quantum statistical mechanics or yet-to-be-discovered statistical mechanics, depending on the model. In all cases the thermodynamic calculus is the same, only the enumeration of microstates—that is, *W*— depends on the physical model.

### 4.2. What is W?

Once the selection functional *W* is specified the most probable distribution is fixed and all canonical variables become known functions of $\bar{x}$. But what *is W*? The selection functional is a placeholder for our knowledge, hypotheses and model assumptions about the stochastic processes that gives rise to the probability distribution of interest. This knowledge fully specifies the distribution. The opposite is not true: given distribution *f* there is an infinite number of functionals *W* that produce that distribution as the most probable distribution in their probability space. This nonuniquness is a feature, not a bug: it allows models that are quite different in their details to produce the same final distribution. Here is an example. The unbiased functional W[h]=w(x)=1 produces the exponential distribution
(43)$$h^{*}\left(x\right)=\frac{e^{-\beta x}}{q},$$
with canonical parameters
(44)$$\beta =1/\bar{x},\;q=\bar{x},\;\log\omega =1+\log\bar{x}.$$

Now consider the nonlinear selection functional
(45)$$\log W\left[h\right]=S\left[h\right]=-\int_{0}^{\infty }h\left(x\right)\log h\left(x\right)dx,$$
whose logarithm is equal to entropy. The corresponding microcanonical probability functional is obtained by inserting this into Equation (Equation 27),
(46)$$\log\varrho \left[h|W,\bar{x}\right]=-2\int_{0}^{\infty }h\left(x\right)\log h\left(x\right)dx-\log\omega$$
and is maximized by (see Appendix A)
(47)$$h^{*}\left(x\right)=w\left(x\right)\frac{e^{-\beta x}}{q},$$
with
(48)$$w\left(x\right)=\bar{x}e^{x/\bar{x}},\;\beta =2/\bar{x},\;q={\bar{x}}^{2},\;\log\omega =2+2\log\bar{x}.$$

We have arrived at the *exponential* distribution, the same distribution that is obtained by the unbiased functional w(x)=1, but with different canonical parameters because the probability space from which it arises is now different. If all we know is that the probability distribution in a stochastic process is exponential, it is not possible to determine whether it was obtained using W[h]=1, $W\left[h\right]=e^{S[h]}$, or any other functionals that is capable of reproducing the exponential distribution. While the selection bias identifies the most probable distribution uniquely, the opposite is not true.

The selection functional represents external input to thermodynamics and is fixed by the rules that govern the stochastic process that produces the distribution in question. In the case of statistical mechanics it is fixed by the postulate of equal a priori probabilities. In another example, recently given for stochastic binary clustering, it is fixed by the aggregation kernel, a function that determines the aggregation probability between clusters of different sizes [17]. The selection functional is the contact point between generalized statistical thermodynamics—a mathematical theory for generic distributions—and *physics*, i.e., our knowledge in the form of model assumptions and postulates about the process that gives rise to the observed distribution. It is interesting to point out that the variational derivative *w* in Equation (Equation 27) appears in the form of Bayesian prior [18]. In the context of generalized thermodynamics *w* is not a prior distribution—although it might if $a_{0}=a_{1}=0$ in Equation (Equation 41). In general, *w* is a non normalizable derivative of the functional that represents our knowledge about the process, an improper prior that points nonetheless to a proper distribution.

## 5. Thermodynamic Sampling of Distributions

We have shown that any distribution f(x) defined in $R_{+}$ can be viewed as the most probable distribution in an appropriately constructed probability space. Here we will show that any distribution *f* in this domain can be obtained as the equilibrium distribution of reacting clusters under an appropriately constructed equilibrium constant. Consider a population of *M* identical particles (“monomers”) distributed into *N* clusters and let $m=(m_{1},m_{2}\cdots ,m_{N})$ be an ordered list of *N* cluster masses with total mass *M* such that $m_{k}$ is the mass of the *k*th cluster in the list (“configuration”). The complete set of configurations with *N* clusters and total mass *M* comprises the cluster ensemble (M,N). Let $n=(n_{1},n_{2}\cdots )$ be the size distribution of the clusters in configuration m such that $n_{i}$ is the number of clusters with *i* monomers. With M,N→∞ at fixed $M/N=\bar{x}$, the cluster ensemble contains every discrete distribution $h_{i}=n_{i}/N$ with mean $\bar{x}$. We now construct the following stochastic process: given a configuration m, pick two clusters at random, merge them, then split them back into two clusters at random. This amounts to an exchange of mass between two clusters that is represented schematically by the reaction
(49)$$m_{i}+m_{j}\overset{\;}{\to }m_{i}^{\prime }+m_{j}^{\prime }$$
and transforms the parent configuration m into an new configuration $m^{\prime }$ with the same number of clusters *N* and total mass *M*. This process may also be represented as a reaction that transforms a parent configuration into an offspring,

(50)$$m\overset{K}{\to }m^{\prime }.$$

We define the equilibrium constant of this reaction as
(51)$$K_{m\to m^{\prime }}=\frac{W(n^{\prime })}{W(n)},$$
where $n^{\prime }$ and n are the cluster size distributions of the product and reactant configurations, respectively, and W(n) is the selection functional applied to distribution n. The change δn of the corresponding distributions upon the exchange reaction is a change of −1 in the number of cluster masses $m_{i}$ and $m_{j}$ on the reactant side, and +1 for cluster masses on the product side. By virtue of the homogeneous property of logW, its change for large *M* and *N* is a differential that can be expressed in terms of the derivatives of logW
(52)$$\log W\left(n^{\prime }\right)-\log W\left(n\right)=-\log w\left(m_{i}\right)-\log w\left(m_{j}\right)+\log w\left(m_{i}^{\prime }\right)+\log w\left(m_{j}^{\prime }\right),$$
where logw is the functional derivative of logW evaluated in distribution n. Using this result the equilibrium constant becomes

(53)$$K_{m\to m^{\prime }}=\frac{w\left(m_{i}^{\prime }\right)w\left(m_{j}^{\prime }\right)}{w\left(m_{i}\right)w\left(m_{j}\right)}.$$

This has the standard form of an equilibrium constant for the reaction in Equation (Equation 49). We may identify w(x) as the “fugacity” of species *x* and “species” as a cluster with mass *x*. The reaction can be simulated by Monte Carlo using the Metropolis transition probabilities
(54)$$P_{n\to n^{\prime }}=\left\{\begin{array}{cc} \mathrm{rnd} & \mathrm{if}\;\mathrm{rnd}\leq K_{n\to n^{\prime }},\\ 1 & \mathrm{if}\;\mathrm{rnd}>K_{n\to n^{\prime }}, \end{array}\right.$$
where rnd is a uniform random number in (0,1). This forms a reducible Markov process that samples the microcanonical space of distribution n with fixed zeroth order moment *N* and first moment *M*. Its stationary distribution is [19]
(55)$$h^{*}\left(x\right)=w\left(x\right)\frac{e^{-\beta x}}{q}.$$
where logw(x) is the functional derivative of logW evaluated at $h=h^{*}$ and the parameters β and *q* are obtained by solving the set of equations

(56)$$\begin{array}{c} q=\int_{0}^{\infty }w\left(x\right)e^{-\beta x}dx, \end{array}$$

(57)$$\begin{array}{c} \bar{x}=\frac{1}{q}\int_{0}^{\infty }xw\left(x\right)e^{-\beta x}dx. \end{array}$$

With W[h]=w[x]=1 we obtain the exponential distribution, which implies that the exchange reaction with equilibrium constant K=1 for all transitions is equivalent to unbiased sampling from an exponential distribution with fixed mean $\bar{x}=M/N$.

Once the selection functional *W* is given the most probable distribution is fixed and may be obtained either by simulation or in many cases analytically. We will now construct *W* such that the most probable distribution is any distribution *f* defined in $R_{+}$. We construct the linearized selection functional
(58)$$\log W\left[h\right]=\int_{0}^{\infty }h\left(x\right)\log w\left(x\right)dx$$
with *w* from Equation (Equation 41), which we write in the form
(59)$$w\left(x\right)=f\left(x\right)e^{a_{0}+a_{1}x}$$
and $a_{0}$ and $a_{1}$ arbitrary constants. It is easy to show that the selection of $a_{0}$ and $a_{1}$ is immaterial because both constants drop out of Equation (Equation 53). If we choose $a_{0}=a_{1}$, then w(x)=f(x); alternatively, we may choose these constants so as to obtain simpler forms for w(x). We demonstrate the construction of *w* with three examples using the exponential, the Weibull, and the uniform distribution.

Exponential distribution
(60)$$f\left(x\right)=e^{-x/\bar{x}}/\bar{x}.$$The function *w* is
(61)$$w\left(x\right)=\frac{e^{-x/\bar{x}+a_{0}+a_{1}}}{\bar{x}}.$$Choosing $a_{0}=\log\bar{x}$, $a_{1}=1/\bar{x}$ we obtain $w_{\exp}\left(x\right)=1$, which represents the unbiased selection functional.Weibull distribution
$$f\left(x\right)=\left(\frac{k}{\lambda }\right){\left(\frac{x}{\lambda }\right)}^{k-1}e^{-{\left(x/\lambda \right)}^{k}}.$$Using $a_{0}=k\log\lambda -\log k$ and $a_{1}=0$ in Equation (Equation 59) we obtain
(62)$$w_{\mathrm{Weibull}}\left(x\right)=x^{k-1}e^{-{\left(x/\lambda \right)}^{k}}.$$Uniform distribution
(63)$$f\left(x\right)=\left\{\begin{array}{cc} 1/(b-a) & a\leq x\leq b\\ 0 & \mathrm{otherwise}. \end{array}\right.$$With $a_{0}=a_{1}=0$ we obtain
(64)$$w_{\mathrm{uniform}}\left(x\right)=f\left(x\right).$$

We implement thermodynamic sampling using Monte Carlo. We begin with an ordered list of *N* integers i>0 whose sum is *M*. We then pick two numbers at random and implement a random exchange reaction to produce a new pair of integers with the same combined sum. The new pair replaces the old with acceptance probability computed according to Equation (Equation 54) using $K_{\mathrm{eq}}$ from Equation (Equation 53) and the function w(x) obtained above. Following a successful trial we calculate the distribution of the current configuration. The mean distribution is obtained by averaging over a large number of trials. For these simulations N=100, M=3000, $\bar{x}=30$, and the mean distribution is calculated over 20,000 trials. As we discuss elsewhere, the mean distribution and the most probable distribution converge to each other unless the system exhibits phase separation [17,19,20]. The results in Figure 1 make it clear that thermodynamic sampling converges indeed to the distribution for which the *w* function was derived. Any discrete distribution $h_{i}$, and with proper scaling, any continuous distribution h(x), may be associated with the equilibrium distribution of reacting clusters under a suitable equilibrium constant.

The selection functionals constructed by the procedure discussed here apply the variational derivative at *f* to all distributions *h*, i.e., they are linearized at the most probable distribution. Any nonlinear functional logW with the same derivative at h=f will produce the same distribution as the stationary distribution under exchange reactions. One example is the entropic functional in Equation (Equation 45), a nonlinear functional that produces the exponential distribution. Even though the entropic and unbiased functionals both produce the same distribution (Figure 2a), their corresponding ensembles are distinctly different because each functional assigns different probabilities to the distributions of the ensemble. This difference can be seen in the fluctuations (Figure 2b). The entropic functional is more selective than the unbiased, which picks configurations with equal probability. Accordingly, fluctuations in the entropic ensemble have narrower distribution. This can be clearly seen in Figure 2b that shows the fluctuations in the number of monomers for the entropic and the unbiased functionals.

## 6. Conclusions

Stripped to its core, what we call statistical thermodynamics is a mapping between a probability distribution *f* and a set of functions, {w,β,q,ω} from which the distribution may be reconstructed. What we call classical thermodynamics is the set of relationships among $\left\{\beta ,q,\omega ,\bar{x}\right\}$—relationships that are the same for all distributions. What we call second law is the variational condition that identifies the most probable distribution in the domain of feasible distributions. What we call quasistatic process is a path in the space of distributions under fixed *W*. Physics enters through *W*. This generic mathematical formalism applies to any distribution. To use an analogy, thermodynamics is a universal grammar that becomes a language when applied to specific problems. It is a fitting coincidence—or perhaps an inevitable consequence—that it was the human desire to maximize the amount of useful work in the steam engine that would eventually make contact with the variational foundation of thermodynamics. Gibbs’s breakthrough was to connect thermodynamics to a probability distribution, and that of Shannon and Jaynes to transplant it outside physics. In the time since, the vocabulary of statistical thermodynamics has felt intuitively familiar across disciplines in a déjà vu sort of manner, even as its grammar remained undeciphered. This intuition can now be understood: The common thread that runs through every discipline that has adopted the thermodynamic language is an underlying stochastic process, and where there is probability, there is statistical thermodynamics.

## Figures and Tables

**Figure 1 entropy-21-00890-f001:**
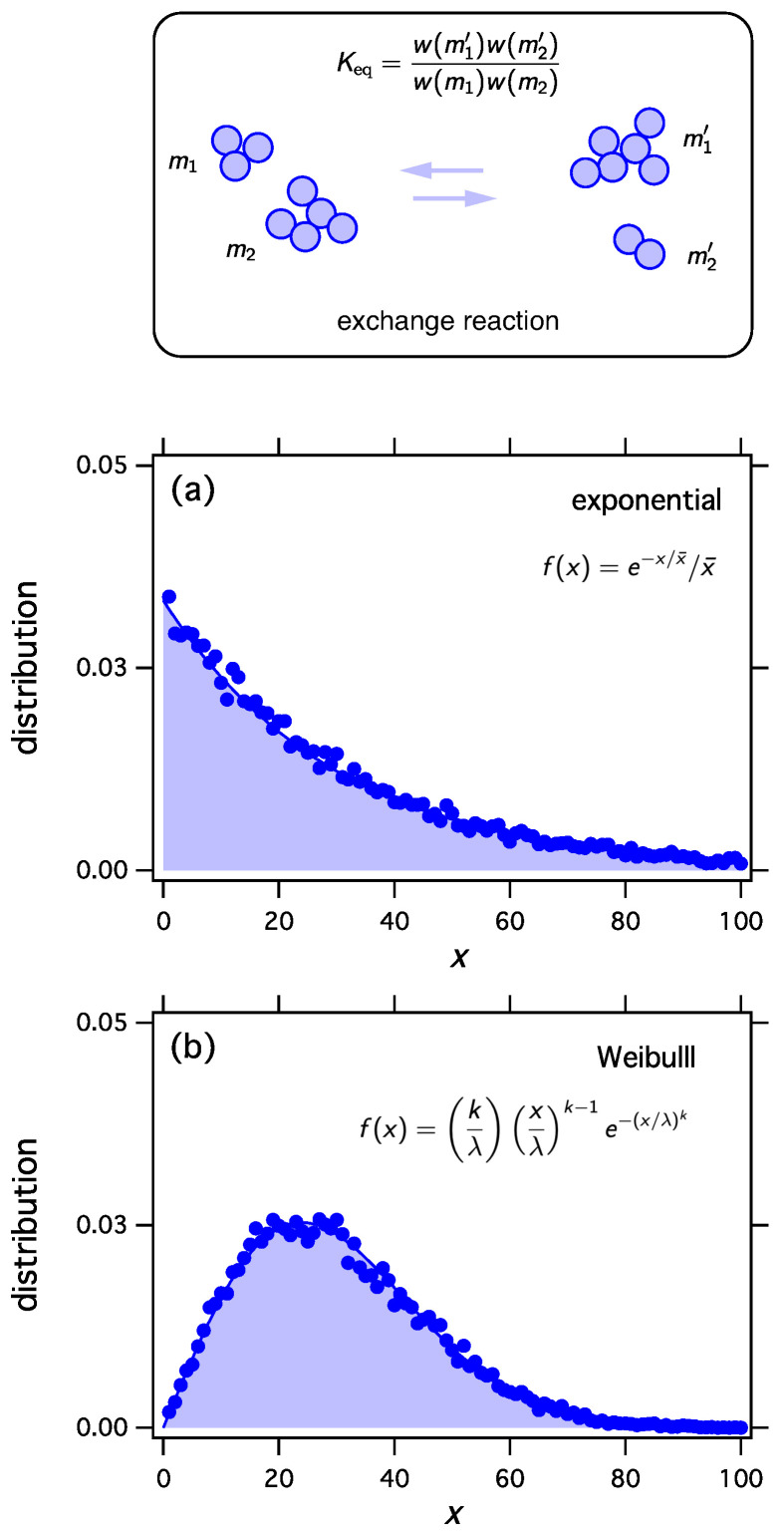
The exchange reaction transfers mass between two clusters and samples the space of all distributions with fixed number of clusters *N* and fixed total number of monomers *M*. We may construct the equilibrium constant of this reaction so as to to obtain any desired equilibrium distribution. Any distribution f(x), x≥0, may be obtained as the equilibrium distribution. In this example we obtain (**a**) the exponential distribution; (**b**) the Weibull distribution with λ=33.8514, k=2; and (**c**) the uniform distribution between a=20 and b=40. In all cases $\bar{x}=30$.

**Figure 2 entropy-21-00890-f002:**
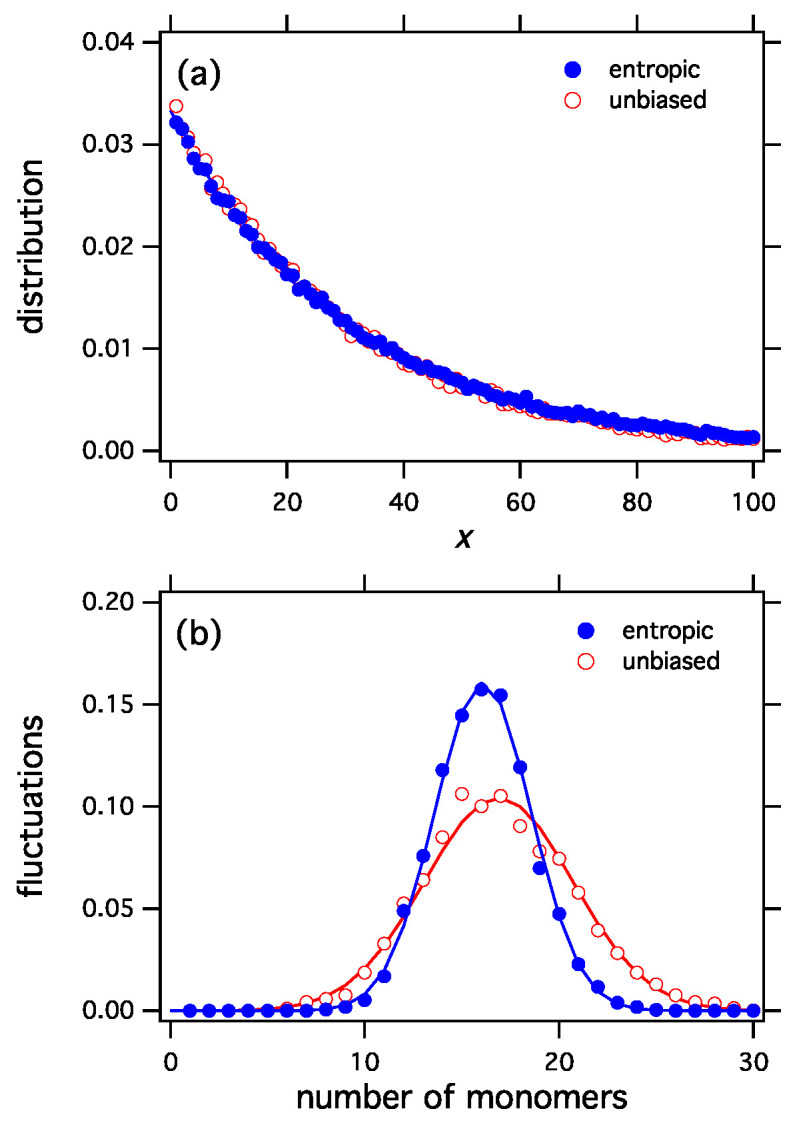
(**a**) The entropic selection functional, $W\left[h\right]=e^{S[h]}$, and the unbiased functional, W[h]=1, both produce the same equilibrium distribution (exponential). Nonetheless the two selection functionals represent distinctly different ensembles, as can be seen in fluctuations of the number of monomers (**b**). The entropic functional is more selective than the unbiased and produces a tighter distribution of fluctuations.

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
