# Peer review of "Thermodynamics Beyond Molecules: Statistical Thermodynamics of Probability Distributions"

_entropy, 2019, doi:10.3390/e21090890_

Round 1
Reviewer 1 Report
The work is an excellent way of synthesizing and formalizing how notions about entropy, thermodynamics and stochastic processes are related in a generic way. The combination of these ideas with the experimental evidence of phenomena that could be analyzed with this framework (biological, physical, cognitive), in a wide variety of systems, including all possible biological scales, is huge, from individual biological systems to populations. The
The applicability of the concepts presented here makes this an exciting time. Ideas that previously seemed tremendously speculative, thanks to work like this, are constituting a powerful framework independent of the analysis of real data on many different systems, and the common characteristics observed at so many levels of biological organization encourage us to think that there really are general principles governing the function of complex systems and that work such as this is a help in this direction.
I support acceptance of the paper although authors should review some typos that appear in the supplementary paper part, along with missing numbering in several equations.
Reviewer 2 Report
This paper tackles an interesting question: what is the scope of thermodynamics? The paper argues that thermodynamics can be generalised. Overall, the paper is original and interesting, but there is one feature that needs to change before I can recommend accpetance.
Throughout the paper, the author does not distinguish between thermodynamics and statistical mechanics. Of course, if one holds there is a reduction of thermodynamics to statistical mechanics then the two are closely connected, but this is a controversial philosophical issue. Thus, the author often write `thermodynamics' when they intend 'statistical mechanics'. Thermodynamics a la Kelvin and Clausius concerns heat and work, and the parameters of a small set of macrovariables, not probability distributions nor microstates - that is the subject matter of statistical mechanics.
Please rectify this throughout but especially in the conclusion, where you discuss the second law as variational condition. Of course, perhaps this variational condition is related to the Kelvin, Clausius or Carnot statements of the second law - but since you have not connected your paper to work and heat, this has not been established here. You end by saying that where there is probability there is thermodynamics - but really you mean there is statistical mechanics. line 145: this expression is the Boltzmann entropy, not the thermodynamic entropy.- There is a typo on line 43, you mean Q not E_i.
